# Factors Influencing Dementia Care Competence among Care Staff: A Mixed-Methods Systematic Review Protocol

**DOI:** 10.3390/healthcare12111155

**Published:** 2024-06-06

**Authors:** Jinfeng Zhu, Jing Wang, Bo Zhang, Xi Zhang, Hui Wu

**Affiliations:** 1Faculty of Nursing, Health Science Center, Xi’an Jiaotong University, Xi’an 710061, China; zhujinfeng@stu.xjtu.edu.cn (J.Z.); zhangbo1997@stu.xjtu.edu.cn (B.Z.); zhangxi_2022@stu.xjtu.edu.cn (X.Z.); 2Ankang Hospital of Traditional Chinese Medicine, Ankang 725000, China

**Keywords:** dementia, competence, care staff, systematic review, mixed-methods

## Abstract

Background: Dementia care competence is defined as the ability, acquired through practical experience, to deliver high-quality care services to persons with dementia (PWD). However, many studies only focus on one aspect of competence using qualitative or quantitative research design and have small sample sizes of care staff with dementia. This study aims to conduct a mixed-methods systematic review of the factors influencing the competence of dementia care staff, and explore the relationship between these factors and competence. Methods: This review was designed following the PRISMA-P 2015 statement and methodological guidance for the conduct of mixed-methods systematic reviews from the Joanna Briggs Institute (JBI). Seven English and four Chinese databases will be searched to systematically review the existing eligible studies. JBI Critical Appraisal Checklist for Qualitative Research and Analytical Cross-Sectional Studies will be used to assess the methodological quality of each study. A JBI Mixed-Methods Data Extraction Form will be applied for data extraction. The JBI convergent integrated approach will be used for data synthesis and integration. The synthesized findings will be graded according to the JBI ConQual approach as high, moderate, low, or very low. The protocol was registered with PROSPERO in October 2023 (CRD42023474093).

## 1. Introduction

Dementia refers to significant cognitive decline in areas such as complex attention, executive function, learning and memory, language, perceptual-motor, or social cognition. This decline interferes with an individual’s independence in everyday activities and behavior changes, necessitating assistance with complex tasks [1,2]. At present, 47.5 million people worldwide suffer from dementia, and the number of cases of dementia is expected to soar to 135.5 million by 2050 [3]. The global aging population and the increasing incidence and prevalence of dementia have led to a growing demand for healthcare services across various settings, whether in hospitals, nursing homes, communities, or households [4,5]. Care staff, comprising healthcare professionals providing medical care or services to individuals in need of health support, play a critical role in dementia care. These professionals, including nurses, nursing/care assistants or aides, and physicians, work across diverse healthcare settings, each requiring distinct competence [6]. Competence is increasingly recognized as an important concept in international research related to this population. Currently, dementia care competence is defined as the ability, acquired through practical experience, to deliver high-quality care to persons with dementia (PWD) [7,8]. Research indicates that higher levels of care staff competence are associated with positive care outcomes for PWD. These outcomes encompass reduced agitation and aggression, decreased use of physical restraints and antipsychotic medications, and an overall enhancement in the quality of life and safety of PWD [9,10,11]. Furthermore, competent care staff can optimize resource utilization, support caregivers, and contribute to their satisfaction and retention [12]. Therefore, understanding the concept of dementia care competence and the factors influencing the competence of dementia care staff is crucial for ensuring optimal care and positive outcomes for PWD. Moreover, by considering these different competence and the factors that influence them, healthcare organizations and policymakers can develop targeted interventions and support systems to enhance dementia care competence among care staff.

In summary, dementia care competence is a multifaceted concept that encompasses various domains crucial for providing comprehensive dementia care [13]. Several individual factors have been identified to positively influence dementia care competence, including educational level, training times, and work experience [14,15]. Professional knowledge forms a solid foundation for dementia care competence, with nurses possessing a deeper understanding of dementia better equipped to deliver high-quality care [16,17]. It is found that nurses with a deeper understanding of dementia and mental diseases are better equipped to provide high-quality care [16]. Moreover, a person-centered attitude is associated with competence in providing dementia care and is also related to job satisfaction [18]. Technical skills also play a crucial role, as demonstrated by studies showing that competency-based skills in activities of daily living care and medical nursing contribute to improved outcomes for PWD [19]. Additionally, due to the decline in cognitive function of PWD, being able to effectively communicate with them, understand their needs and desires, and provide emotional support and reassurance is crucial. Facilitating good communication and cooperation between PWD, their family members, and other team members is essential for dementia care staff to be more competent in their work [20]. Continuous learning and updating knowledge, and actively participating in training and further education, can positively influence care staff confidence in managing behavior changes in PWD [21]. The structural inadequacies of an acute hospital create numerous challenges for care staff in caring for PWD, adding complexity to the care experience, which highlights the impact of the environment on dementia care competence [22]. These factors interact with each other, collectively influencing the level of competence among care staff. A framework for dementia care competence provides a structured approach to understanding and assessing the necessary skills and attributes required by care staff. A scoping review outlined a comprehensive framework for dementia care competence, identifying 13 principles competencies to organize dementia care competence for care staff, including promoting health and social well-being, identifying dementia—know the early signs, assess and diagnose, and several others [23]. Despite the existing research highlighting the association between higher care staff competence and positive health outcomes for PWD, previous studies mainly focused on a single aspect of competence, with a small sample and a lack of comprehensive research. Consequently, researchers frequently overlook influential factors when designing studies aimed at enhancing the competence of dementia care staff. As of now, a conclusive consensus on the factors influencing dementia care competence among care staff remains elusive, and the relationship between various influencing factors and their impact on competence remains unknown.

Quantitative research solely focuses on assessing whether care staff competence has improved, with limitations in capturing care staff’s feelings, needs, and changes at different stages [11]. Qualitative studies have delved into care staff experiences in dementia care from various perspectives, but a certain degree of subjectivity remains and the sample sizes are relatively small [24]. Neither quantitative nor qualitative research in isolation can provide a comprehensive understanding of care staff competence through a single study. Systematic reviews are widely recognized as the highest level of evidence and are extensively used in clinical practice. Currently, most systematic reviews in dementia care competence are based on quantitative research findings, with only a few incorporating qualitative research results. Nevertheless, whether based on quantitative or qualitative research, systematic reviews essentially represent reviews of single research methodologies. In certain contexts, their applicability is limited, challenging the provision of comprehensive and viable recommendations for both frameworks and practices [24]. A mixed-methods systematic reviews combine evidence from quantitative, qualitative, and mixed methodological studies together and contribute to an in-depth understanding of complex phenomena, interventions, and programs in the field of public health [25,26]. They aim to maximize the findings and the ability of those findings to inform policy and practice, increasing the usefulness of supporting decision makers in the healthcare environment [25]. To the best of our knowledge, no existing mixed-methods systematic review on this topic has been identified.

This study aims to conduct a mixed-methods systematic review of the factors influencing the competence of dementia care staff, and explore the relationship between these factors and competence. The findings from this study will provide evidence about factors correlated with dementia care competence among care staff, and inform the development and implementation of dementia care training programs targeting care staff’s needs. This study also has broader applicability to those interested in workforce education and training.

## 2. Materials and Methods

This protocol was designed following the Preferred Reporting Items for Systematic Reviews and Meta-Analysis Protocols (PRISMA-P) 2015 statement [27,28] and methodological guidance for the conduct of mixed-methods systematic reviews from the Joanna Briggs Institute (JBI) [24]. The PRISMA-P checklist is available in the online Appendix A [27]. The protocol was registered with the International Prospective Register of Systematic Reviews (PROSPERO) in October 2023, and the registration number is CRD42023474093. The study will run from October 2023 to August 2024.

### 2.1. Identification of the Search Question

The main question of the study is: What are the factors influencing dementia competence among care staff? What is the relationship between these factors and dementia competence among care staff?

### 2.2. Eligibility Criteria

#### 2.2.1. Study Design

This study design applies both qualitative and quantitative primary studies.

Qualitative studies: The search will include, but is not limited to phenomenology, grounded theory, ethnography, or narrative approaches. The qualitative search criteria will be guided by the Population, Phenomenon of Interest, Context (PICo) framework [29] (see Table 1).Quantitative studies: The search will focus on cross-sectional studies exploring the factors influencing dementia care competence among care staff. The quantitative search criteria will be guided by the Population Exposure Comparator Outcome (PECO) framework [30] (see Table 1).Mixed-methods studies: Studies that comprise both qualitative research and cross-sectional studies will also be included.

Secondary studies such as scoping reviews, narrative reviews, systematic reviews, and pilot studies will be excluded.

#### 2.2.2. Participants and Outcome

Quantitative component: This review will consider studies that include outcome measures related to the competence of dementia care staff. The primary outcome of the study focuses on the factors influencing dementia care competence among care staff. This may encompass self-assessment of competence, levels of dementia-specific professional knowledge, attitudes, skills, judgment, challenge in caring for PWD, actual caregiving behaviors, and other relevant measures of competence.Qualitative component: This review will consider studies that include care staff experience and perception about their competence in caring for PWD in hospitals, nursing homes, or the community.

#### 2.2.3. Languages

Studies published in languages other than Chinese or English will be excluded.

**Table 1 healthcare-12-01155-t001:** Eligibility criteria using the PECO and PICO format.

		Inclusion Criteria	Exclusion Criteria
	(P) Population *	All healthcare professionals who provide medical care or services to individuals in need of health support. They include but are not limited to nurses, nursing/care assistants or aides, and physicians.	Nursing students. Family caregivers. Persons with dementia.
Qualitative data	(I) Phenomena of interest	Care staff experience and perception of competence in caring for persons with dementia.	Studies that address general healthcare competence without specific emphasis on dementia care. Studies that focus on aspects other than competence, such as general experience towards unrelated healthcare practices, ethnic minority issues, and ethical problems unrelated to dementia care.
	(Co) Context	Includes but is not limited to hospital, nursing home, or the community	Home-based care provided by family members. Any non-healthcare settings, such as administrative offices or purely academic environments.
Quantitative data	(E) Exposure	Cross-sectional study that analyzes factors influencing dementia care competence among care staff.	Studies focusing solely on interventions or educational programs without assessing competence directly.
	(C) Comparator	Not applicable.	Not applicable.
	(O) Outcome	The primary outcome of the study focuses on the factors influencing dementia care competence among care staff. This may encompass self-assessment of competence, dementia-specific professional knowledge, attitudes, skills, judgment, challenge in caring for persons with dementia, actual caregiving behaviors, and other relevant measures of competence.	Studies that measure outcomes unrelated to dementia care competence, such as general job satisfaction or burnout.

Note: * The column labeled “(P) Population” is used for both quantitative and qualitative data.

### 2.3. Information Sources

Seven English and four Chinese databases will be searched to systematically review the existing eligible studies. Searches on English databases like PubMed, Medline, Web of Science Core Collection, Embase, Scopus, Cochrane Library, and CINAHL will be carried out. The following Chinese databases will also be consulted for relevant data: China National Knowledge Infrastructure, China Science and Technology Journal Database, Wanfang Database, and Chinese Biomedicine Literature Database. There are no restrictions regarding the publication period. Research published in English and Chinese will be considered. Reference lists and citations will be manually searched to ensure a thorough search.

### 2.4. Search Strategy

By searching and reading relevant published papers related to the concept of competence, the scope of the previous literature is determined to ensure familiarity with the topic and to generate search terms. The Booleans ‘OR’, ‘AND’, title, abstract, and keywords will be used for comprehensive searching. Preliminary searches will be carried out to validate the sensitivity and specificity of the approach across all databases prior to initiating the formal search. The final strategy will be formulated by scoping the previous literature and pre-search with an expert librarian and our research group. We will incorporate research accessible from the establishment of the database up to 24 May 2024. Appendix A shows the descriptions of each database.

### 2.5. Study Selection

Following the search, all identified citations will be loaded into EndNote 21.2 (Clarivate Analytics, Philadelphia, PA, USA) and duplicates will be removed. First, irrelevant papers will be excluded through the title and abstract. Second, the full text will be retrieved for further screening. We will contact the study author if more information is needed to determine eligibility or if there is difficulty in obtaining the full text. The full text of selected studies will be assessed in detail against the inclusion criteria by two independent reviewers (J.Z. and B.Z.). Reasons for excluded studies that do not meet the inclusion criteria will be recorded and reported in the systematic review. Any disagreements that arise between the reviewers at each stage of the study selection process will be resolved through group discussion, or with a third reviewer (X.Z.). The results of the search will be reported in full in the final review, and the process of study selection will be presented in a Preferred Reporting Items for Systematic Reviews (PRISMA) flow diagram in Figure 1 [31].

### 2.6. Assessment of Methodological Quality

JBI Critical Appraisal Checklist for Analytical Cross-Sectional Studies [32] and Qualitative Research [29] will be used to assess the methodological quality of each study. Mixed-method study will be separately evaluated, with the quantitative component being assessed by JBI Critical Appraisal Checklist for Analytical Cross-Sectional Studies, and the qualitative component of mixed-methods studies being evaluated by JBI Critical Appraisal Checklist for Qualitative Research. JBI Critical Appraisal Checklist for Analytical Cross-Sectional Studies contains 8 items and JBI Critical Appraisal Checklist for Qualitative Research uses 10 items. Each item is rated as “yes”, “no”, “unclear”, or “not applicable”.

Following critical appraisal, studies that do not meet a certain quality threshold will be excluded [33]. The authors of the articles will be contacted to request missing or additional data in order to obtain clarification, when required. Two researchers (J.Z. & B.Z.) will appraise the methodological quality of each article independently. Prior to each formal procedure, a pilot examination will be carried out initially with 5–10% of studies scrutinized independently by all reviewers. This aims to enhance comprehension and implementation of eligibility criteria and the assessment tool. Disagreements will be solved by the discussion with a third reviewer or within the study group till a consensus is reached. The results of critical appraisal will be reported in narrative form and presented in a table.

### 2.7. Data Extraction

JBI Mixed-Methods Data Extraction Form following a Convergent Integrated Approach will be applied for data extraction [34]. Our research team has adapted the tool based on the needs of this study. For example, we increase the data item of ‘objectives’ to learn about the aim of the study and ‘operationalization of sense of competence’ to describe the competence measure method and tool. We will use 5–10% of the included studies to do a pilot extraction. Before initiating the formal data extraction process, data from 5% to 10% of the included studies will be extracted using the data extraction tool to assess the tool’s effectiveness and mitigate the possibility of errors. Training of the researchers (J.Z. and B.Z.) involved in data extraction will also be conducted to confirm the consistency. Data will be critically reviewed and extracted by two researchers (J.Z. and B.Z.) independently after a consensus meeting. If there are any disagreements regarding data extraction, the research group will negotiate until a consensus is reached. The authors of the articles will be contacted to request missing or additional data in order to obtain clarification, when required.

The extracted data will include detailed descriptions of selected studies such as author, methods, settings, participants, samples, objectives, and key findings. The qualitative data extracted will include the illustration within themes and sub-themes related to the research question. Then, each finding will be assigned a level of credibility determined by the congruence of the finding with supporting data. The credibility is delineated across three levels: unequivocal, credible, and not supported. The unequivocal level signifies evidence established beyond a reasonable doubt, whereas the credible level denotes findings deemed plausible within the theoretical framework and substantiated by empirical data. Conversely, if the evidence lacks empirical substantiation, it is assigned the category of not supported. The quantitative data extracted entail outcomes grounded in data pertinent to the research questions [34]. Quantitative data will comprise data-based outcomes of descriptive and/or inferential statistical tests. The study findings should include all factors that are relevant to the competence among dementia care staff, both significant and non-significant results. Table 2 shows the details of the data extraction codebook.

### 2.8. Data Transformation

The data will be transformed into a mutually compatible format in order for qualitative and quantitative data to be integrated and fully inform the topic [35]. Data transformation includes converting qualitative data into quantitative data (i.e., quantitizing) or converting quantitative data into qualitative data (i.e., qualitizing) [24]. By qualitizing, the reviewer converts the quantitative data into qualitized data. These qualitized findings are then assembled and pooled into categories with qualitative findings extricated directly from qualitative studies. Qualitizing will be recommended to conduct a mixed-methods review within a convergent integrated approach. This approach is favored over attributing numerical values to qualitative data, as it reduces the susceptibility to errors in coding quantitative data and helps mitigate bias. Furthermore, the analysis of qualitative data plays a crucial role in complementing quantitative data in supplementary research, contributing to a better understanding of the influence of contextual factors.

### 2.9. Data Synthesis and Integration

Considering the nature of the review question in the present systematic review, the convergent integrated approach will be applied. Thematic synthesis will be applied to the integration of findings when quantitative data are qualitized. There are six phases in thematic synthesis, which include familiarizing with data, generating initial codes, searching for themes, reviewing potential themes, defining and naming themes, and producing the report [36].

In phases “familiarizing yourself with your data”, the reviewers need to read the data several times to be familiar with included studies. During the reading process, some ideas or early codes will be formed in our minds. In phases “generating initial codes”, the reviewers look for similarities and differences between the codes in order to start grouping them into a hierarchical tree structure. New codes will be created to capture the meaning of groups of initial codes. In phases “searching for themes”, this stage begins with analyzing the codes and considering the linkable relationship between codes, sub-themes, and themes. The reviewers need to organize and summarize the existing codes under different potential themes. They can draw the logical relationships between different codes, sub-themes, and themes by drawing or using Mind maps/Concept maps in NVivo 12 Plus software to sort out the logical relationships between different codes. In phases “reviewing potential themes”, the reviewers review the coded data to assess whether they appear to exhibit a coherent pattern and consider the validity of individual themes to ascertain whether they accurately capture the meaning evident within the entire data set [37]. At this stage, the reviewers may require various adjustments within the initial coding and themes. In phases “defining and naming themes”, the reviewers will define and name themes, and consider the data in each theme fitting into the overall story of the research questions. It is crucial for researchers to have a clear and well-defined understanding of what the themes encompass and what falls outside their scope at the end of this phase. In phases “producing the report”, the reviewers should present the data in a coherent, concise, logical, non-repetitive, and engaging manner [37].

### 2.10. Assessing Confidence in the Findings

To enhance the credibility of claims associated with a data set, researchers are encouraged to effectively communicate the logical development of findings [36]. The synthesized findings will be graded according to the JBI ConQual approach as high, moderate, low, or very low [38]. Dependability and credibility are pivotal factors influencing the confidence in qualitative synthesized findings. Dependability is established by scrutinizing the quality of the original studies included, utilizing a set of critical appraisal questions. Meanwhile, credibility assesses the alignment between the author’s interpretations and the original data. Initially, the confidence in synthesized findings will be presumed to be high. Subsequently, it will be adjusted downwards based on the outcomes of dependability and credibility assessments. Finally, each synthesized finding will be graded as high, moderate, low, or very low. Moreover, we will follow the GRADE guidelines for publication bias assessment, such as using a Funnel Plot [39]. By plotting the effect size of each study (e.g., correlation coefficient, regression coefficient) against its standard error (or the reciprocal of sample size), the Funnel Plot can depict the symmetry and distribution of study results. A symmetric Funnel Plot implies minimal publication bias, while an asymmetric Funnel Plot may indicate substantial publication bias.

## 3. Discussion

Dementia care staff competence is a complex concept that encompasses personal and external, visible and invisible, individual, and environmental factors [40]. The research tries to explore the existing dementia care principles and align new concepts, and subsequently analyze how these factors impact care staff competence from micro-, meso-, and macro-levels. By doing so, this research endeavors to enhance the theoretical framework for dementia care staff competence and link competence development in dementia care to broader system transformation. The research findings serve as a powerful reference for care staff to improve their own competence, guiding them to be better qualified for their jobs and enhance their competitiveness in the workplace. This creates a win–win situation, igniting care staff enthusiasm for their work and enhancing their sense of achievement, while also contributing to improving the quality of medical services for PWD.

Moreover, there are implications for care staff training and education, as well as for healthcare leaders and policymakers, related to potential enablers for effective dementia care. We anticipate using the results of this review to inform the co-design development of a holistic competence framework to guide approaches to dementia care training and education. Given that factors influencing care staff competence operate at meso- and macro-levels, consultation with key stakeholders, including PWD, family caregivers, and health professionals, will be integral in formulating the next research plan. It is anticipated that this future work will develop the framework in detail, focusing squarely on the expected competence that matter most to PWD.

Strengths of this study include its adherence to the Preferred Reporting Items for Systematic Reviews and Meta-Analysis protocols (PRISMA-P) 2015 statement and methodological guidance for conducting mixed-methods systematic reviews from the Joanna Briggs Institute (JBI). The adoption of a consensus-based approach to analysis ensures collective accountability for interpretative decisions, thereby enhancing the robustness and credibility of the findings. Despite the comprehensive approach, a limitation is that a search for grey literature was not conducted, meaning that studies reported in non-traditional publications were not considered.

## 4. Conclusions

Through a mixed-methods systematic review, we will gain comprehensive insights into the diverse factors impacting the competence of dementia care staff. The research contributes to a deeper understanding of the theoretical framework surrounding dementia care staff competence. This guides future research and practice, further refining and developing the theoretical underpinnings of dementia care. Additionally, by identifying key competence elements, we can develop more effective training programs to help care staff enhance their competence levels. Ultimately, this research contributes to improving the quality of healthcare services for PWD. By fostering more competent care staff and providing more systematic training, we ensure PWD receive better care, enhancing their quality of life. This enables nursing facilities and policymakers to better understand how to enhance care staff competence and provide higher-quality care services.

## Figures and Tables

**Figure 1 healthcare-12-01155-f001:**
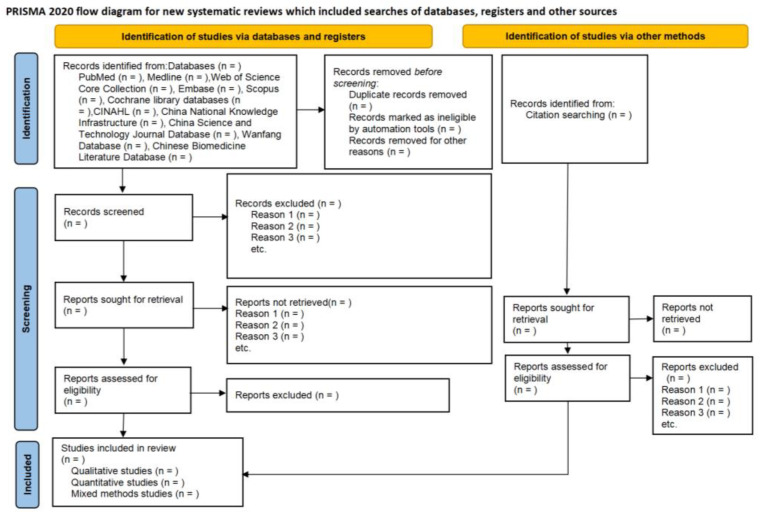
Study selection of the review (from [31]).

**Table 2 healthcare-12-01155-t002:** Data extraction codebook.

	Data Item	Operational Definition
Qualitative data	Author	Author Year
	Design/analysis	Qualitative approach The approach to data collection Qualitative data analysis
	Participants	Sample size Care staff’s characteristics including nurses, care assistants, or nurse managers Care staff’s age, gender, work experience
	Phenomena of interest	The aim of this study
	Country/setting	Country Settings where data collected
	Key findings	Theme and sub-theme Illustrations to support the themes and sub-themes
Quantitative data	Author	Author Year
	Design/analysis	Study design The approach to data analysis
	Objective	The aim of this study
	Sampling method	The method of sampling
	Participants	Sample size Care staff’s age, gender Care staff’s characteristics including nurses, care assistants, or nurse managers
	Operationalization of sense of competence	Competence measure method and tool
	Country/setting	Country Settings where data collected
	Key findings	The results and conclusion of this study

## Data Availability

Data are contained within the article and Appendix A.

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
