# Peer review of "Factors Influencing Dementia Care Competence among Care Staff: A Mixed-Methods Systematic Review Protocol"

_healthcare, 2024, doi:10.3390/healthcare12111155_

Round 1
Reviewer 1 Report
Comments and Suggestions for Authors
The number of patients with dementia is steadily increasing. It is still impossible to find ways to completely cure this pathology. Patients suffering from dementia need constant professional care. Moreover, the number of such patients admitted to hospitals, including for emergency care, is also increasing. Improving the quality of care provided to people with dementia by specially trained staff in hospitals, rehabilitation centres and urgent care centres is an important issue. A systematic mixed methods review protocol to study factors influencing dementia care competence among care staffs will compare knowledge relevant to the areas of search in the conducted studies, analyzing nursing knowledge and experience, the influence of psychological factors, the nursing environment and other factors as they relate to high quality patient care, what favourable outcomes for patients and nursing staff may result. This study protocol is competently written and detailed; however, I have several questions and recommendations for the authors:
1. I recommend splitting the Inclusion and Exclusion criteria column into two (line 111) for comfort understanding of the information reflected in Table 1.
2. I recommend inserting examples in the search strategies that will use keywords, phrases, etc. Line 125.
3. How will you conduct risk of bias and quality assessment?
4. How will you assess publication bias?
5. Since the Results section is not presented in your paper, the question arises: What will be your primary result? What will be the secondary outcomes?
Regards, reviewer.
Reviewer 2 Report
Comments and Suggestions for Authors
Comments
This study investigated the Factors influencing dementia care competence among care staffs. The study, which aim is to provide evidence about factors correlated with dementia care competence among care staffs, and inform the development and implementation of dementia care training programs targeting at care staff’s needs, is very interesting.
Nevertheless, I have some comments and suggestions for authors, listed below:
Major comments
Introduction:
The authors should define dementia, with new criteria proposed by American Psychiatric Association, 2013, DSM-5.
Line 111: Table 1 : The authors will include therapists in the study. This term is a little vague. They could give a few examples.
A conclusion is missing.
Minor comments (comments on the whole manuscript)
A space is required before and after square brackets [ ]. The authors should make this change throughout the manuscript.
Line 111: Table 1: A dota is missing for “outcome”. The authors could write:
· Measuring or assessing dementia care staffs’ compe-
tence, confidence, ability or self-efficacy to deliver
high-quality care services to people with dementia, in-
cluding
• dementia-specific professional knowledge
• dementia-specific professional attitudes
• dementia-specific professional skills.
Line 111: Table 1: Below the table, authors should add the word Note, in italics, followed by a full stop, before specifying the legend.
Line 114: “Five English databases and four Chinese databases…”
The authors could avoid repeating the word "databases”
ð “Five English and four Chinese databases…”
Lines 115-119: “Searches on English databases like PubMed, Web of Science, Embase, Scopus, and Cochrane Library. Chinese databases will be conducted China National Knowledge Infrastructure, China Science and Technology Journal Database, Wanfang Database, and Chinese Biomedicine Literature Database will also be looked into for relevant data.”
The sentences in this paragraph are grammatically incorrect. The authors could write:
ð “Searches on English databases like PubMed, Web of Science, Embase, Scopus, and Cochrane Library will be carried out. The following Chinese databases will also be consulted for relevant data: China National Knowledge Infrastructure, China Science and Technology Journal Database, Wanfang Database, and Chinese Biomedicine Literature Database.”
Lines 128-129: the word “literature” is wrongly divided.
ð “Lite-rature”
Line 145: Figure 1 is difficult to read. The authors could enlarge the diagram
Line 146: the letter S should not be in bold type.
Lines 157-158 and 177-178: “Authors of papers will be contacted to request missing or additional data for clarification where required.” The term “where” is incorrect. The authors could write:
ð The authors of the articles will be contacted to request missing or additional data in order to obtain clarification, when required.”
Line 193: Table 2
To improve readability, it would be better to insert a line between the sub-sections or to create a table with horizontal lines.
Lines 269-278: Strengths and limitations of this study should be written in a single paragraph, integrated into the discussion section.
Line 301: References: Only year of publication has to be in bold type. Journal’s names should not be written in capital letters.
Reviewer 3 Report
Comments and Suggestions for Authors
The authors present a protocol for a proposed systematic review of mixed methods studies evaluating factors influencing dementia care competence among care staff. The protocol has been registered in PROSPERO. The topic is of great public health importance. Suggestions for improvement:
1) Please add a completed PRISMA-protocol (PRISMA-P) checklist with location (page number in your manuscript) for each item mentioned in the checklist. [Link to checklist: https://www.prisma-statement.org/protocols] Reference: Moher D, Shamseer L, Clarke M, Ghersi D, Liberati A, Petticrew M, Shekelle P, Stewart LA. Preferred Reporting Items for Systematic Review and Meta-Analysis Protocols (PRISMA-P) 2015 statement. Syst Rev. 2015;4(1):1. doi: 10.1186/2046-4053-4-1
2. Abstract: lines 9-10, "Competence is an important concept which defines staffs’ feeling of being able to manage the caregiving task." This sentence implies that competence is exclusively self-perceived, which is inaccurate. If your review is restricted to self-perceived competence, then please clarify so in the title; or else please revise this statement in your abstract.
3. I have concerns regarding the sensitivity of your search strategy. Firstly, it looks like you identified only 124 publications on PubMed (information in supplement), which seems like a suspiciously small number. Please try to improve the sensitivity of your search. Some tips to do so are mentioned below in comments 4 & 5.
4. In your search strategy, you have restricted your search to nursing staff (line 2 and line 4 in search strategy); please consider including all direct care staff or else specify in your title that you are only interested in nursing staff as your population of interest.
5. In your search strategy, you have used "competency" as a mandatory key word... please consider removing this as a key word. Competency is not a commonly used key word for indexing all research related to care competency. Therefore, by forcing "competency" as a keyword in your search, you may miss out on important studies related to your topic. This might be a reason why you found only a small number of publications (n=124 ti/ab) on PubMed. For examples of search keywords, please look at the following paper: https://www.ncbi.nlm.nih.gov/pmc/articles/PMC10262338/ Kay, K., Metersky, K., Smye, V., McGrath, C., Johnson, K., Astell, A., Sun, W., & Bartfay, E. (2023). A scoping review to inform the development of dementia care competencies. Dementia (London, England), 22(5), 1138–1163.
6. Introduction is very short and insufficient. Your review is aimed at identifying factors which influence dementia care competency, but you do not sufficiently introduce/discuss dementia care competency itself (e.g. what are the different competencies?). Please add a framework for dementia care competencies and add a list of different dementia care competencies and consider giving examples of factors which might influence these different competencies. Please see the following paper for more information: Kay, K., Metersky, K., Smye, V., McGrath, C., Johnson, K., Astell, A., Sun, W., & Bartfay, E. (2023). A scoping review to inform the development of dementia care competencies. Dementia (London, England), 22(5), 1138–1163. https://doi.org/10.1177/14713012231165568
7. In data sources, please consider adding CINAHL database. Specifically for nursing related research, CINAHL database is more specific and important to include.
Comments on the Quality of English Languagen/a
Round 2
Reviewer 3 Report
Comments and Suggestions for Authors
Thank you for submitting a revised manuscript. I have no additional comments.
Comments on the Quality of English Languagen/a